# Galantamine-Memantine Combination in the Treatment of Parkinson’s Disease Dementia

**DOI:** 10.3390/brainsci14121163

**Published:** 2024-11-21

**Authors:** Emma D. Frost, Swanny X. Shi, Vishnu V. Byroju, Jamir Pitton Rissardo, Jack Donlon, Nicholas Vigilante, Briana P. Murray, Ian M. Walker, Andrew McGarry, Thomas N. Ferraro, Khalid A. Hanafy, Valentina Echeverria, Ludmil Mitrev, Mitchel A. Kling, Balaji Krishnaiah, David B. Lovejoy, Shafiqur Rahman, Trevor W. Stone, Maju Mathew Koola

**Affiliations:** 1Cooper Neurological Institute, Cooper University Health Care, Camden, NJ 08103, USA; 2Department of Neurology, Montefiore Medical Center, Bronx, NY 10467, USA; 3Cooper Medical School, Rowan University, Camden, NJ 08103, USA; 4Chase Brexton Health Care, Baltimore, MD 21201, USA; 5Department of Biomedical Sciences, Cooper Medical School, Rowan University, Camden, NJ 08103, USA; 6Research and Development Department, Bay Pines VAHCS, Bay Pines, FL 33744, USA; 7Medicine Department, Universidad San Sebastián, Concepción 4081339, Bío Bío, Chile; 8Department of Geriatrics and Gerontology, New Jersey Institute for Successful Aging, Rowan-Virtua School of Osteopathic Medicine, Stratford, NJ 08084, USA; 9Department of Neurology, University of Tennessee Health Science Center, Memphis, TN 38163, USA; 10Macquarie Medical School, Faculty of Medicine, Health and Human Sciences, Macquarie University, Sydney, NSW 2113, Australia; 11Department of Pharmaceutical Sciences, College of Pharmacy, South Dakota State University, Brookings, SD 57007, USA; 12Nuffield Department of Orthopedics, Rheumatology and Musculoskeletal Sciences (NDORMS), University of Oxford, Oxford OX3 7LD, UK; 13Department of Public Safety and Correctional Services, Baltimore, MD 21215, USA

**Keywords:** cognition, drug combination, galantamine, kynurenic acid, memantine, *N*-acetylcysteine, neuropharmacology, Parkinson’s disease dementia, Parkinson’s disease treatment

## Abstract

Parkinson’s disease (PD) is a progressive neurodegenerative disorder that affects over 1% of population over age 60. It is defined by motor and nonmotor symptoms including a spectrum of cognitive impairments known as Parkinson’s disease dementia (PDD). Currently, the only US Food and Drug Administration-approved treatment for PDD is rivastigmine, which inhibits acetylcholinesterase and butyrylcholinesterase increasing the level of acetylcholine in the brain. Due to its limited efficacy and side effect profile, rivastigmine is often not prescribed, leaving patients with no treatment options. PD has several derangements in neurotransmitter pathways (dopaminergic neurons in the nigrostriatal pathway, kynurenine pathway (KP), acetylcholine, α7 nicotinic receptor, and N-methyl-D-aspartate (NMDA) receptors) and rivastigmine is only partially effective as it only targets one pathway. Kynurenic acid (KYNA), a metabolite of tryptophan metabolism, affects the pathophysiology of PDD in multiple ways. Both galantamine (α7 nicotinic receptor) and memantine (antagonist of the NMDA subtype of the glutamate receptor) are KYNA modulators. When used in combination, they target multiple pathways. While randomized controlled trials (RCTs) with each drug alone for PD have failed, the combination of galantamine and memantine has demonstrated a synergistic effect on cognitive enhancement in animal models. It has therapeutic potential that has not been adequately assessed, warranting future randomized controlled trials. In this review, we summarize the KYNA-centric model for PD pathophysiology and discuss how this treatment combination is promising in improving cognitive function in patients with PDD through its action on KYNA.

## 1. Introduction

### 1.1. History and Clinical Characteristics

PD is a progressive neurodegenerative disorder characterized by both motor and nonmotor symptoms, including cognitive impairments. While first described by James Parkinson in 1817 as a “shaking palsy” characterized by “involuntary tremulous motion”, this initial report claimed cognition and intellect were unaffected. Among the various nonmotor manifestations, autonomic dysfunction with resultant orthostatic hypotension, abnormal sweating, sphincter, or erectile dysfunction are common with progression and can be disabling in later stages of PD [1]. However, PD patients also demonstrate cognitive decline, termed PD dementia (PDD) [1]. It has been reported that 24–31% of people diagnosed with PD also develop cognitive decline, and patients with PD are four to six times more likely to develop cognitive decline than matched healthy individuals [1]. It affects 75% of those who survive ten or more years after diagnosis [2]. Other neurobehavioral abnormalities include, but are not limited to, depression, apathy, anxiety, hallucinations, and obsessive–compulsive/impulsive behaviors [1]. Additionally, rapid eye movement (REM) sleep disturbances and sensory abnormalities are an inherent part of the disease, which are often some of the first presenting symptoms [1]. Early detection and early treatment can significantly improve quality of life. Early detection allows medication to be started before these manifestations can lead to poor quality of life, hospitalization, and more significant healthcare costs to patients and caregivers [3,4].

### 1.2. Epidemiology and Risk Factors

PD is the second most common neurodegenerative disorder after Alzheimer’s disease (AD). It is characterized pathologically by the loss of nigrostriatal dopaminergic neurons and the appearance of α-synuclein (αSyn)-containing Lewy bodies or Lewy neurites [5]. PD is estimated to affect 0.3% of the general population and to be influenced by a combination of demographic, genetic, and environmental factors. However, increasing age is the most significant known risk factor, as PD affects over 1% of the population over age 60 and up to 4% of those aged 85–94 years [6,7,8]. PD is affected by sex, being more frequently diagnosed in men than women at an approximate ratio of 2:1 [5,9,10]. Although most cases of PD are idiopathic, 10–15% have a genetic predisposition [11].

### 1.3. Current Treatments

There are 17 approved medications on the market to aid movement symptoms. However, they do not ameliorate many of the other disabling nonmotor symptoms [12]. Levodopa (L-dopa), introduced in 1975, remains the most efficacious treatment for the cardinal motor features of PD [13]. Supportive treatment involves the off-label use of antidepressants, anxiolytics, and muscle relaxants [2]. Current treatments related to cognitive impairments primarily focus on symptom management and lifestyle modification to improve quality of life [2]. In 2000, rivastigmine became the first (and is still the only) medication to gain US Food and Drug Administration (FDA) approval for treatment of PDD [13]. It has been proposed as an inhibitor of acetylcholinesterase (AChEI) and butyrylcholinesterase (BuChE), which break down acetylcholine (ACh) in the synaptic cleft [14]. Preclinical studies suggest that inhibition of BuChE is most responsible for the cognitive benefits of rivastigmine [15,16]. When compared to other drugs such as donepezil, rivastigmine shows greater inhibitory potency against brain AChE and BuChE [15]. Preclinical evidence shows that hippocampal ACh levels may increase up to 30–50% more with rivastigmine compared to donepezil [17,18]. This observation led to the emergence of rivastigmine as the predominant therapy for PDD after its safety and modest efficacy were also shown in clinical trials [19,20]. Rivastigmine has been clinically helpful for AD but not for mild cognitive impairment in PD (PD-MCI) [21]. Since the FDA approval of rivastigmine more than two decades, no other drugs have been approved for PDD. Memantine is an N-methyl-D-aspartate (NMDA) receptor antagonist and is considered investigational for PDD use. Galantamine and memantine were FDA approved for cognitive dysfunction in AD in 2001 and 2003, respectively [22,23,24].

## 2. The Kynurenine Pathway

### 2.1. Therapeutic Potential for Cognition

L-Tryptophan (L-TRP) is one of the nine essential amino acids and is a precursor to crucial metabolic pathways in the CNS. It is transported into the CNS across the blood–brain barrier (BBB) via the L-amino acid transporter 1 (LAT-1) [25]. A minority of TRP that enters the brain (~5%) is used to generate proteins, serotonin (5-HT), and melatonin [26]. The kynurenine pathway (KP) is the main route (~95%) for the metabolism of TRP. Intermediate products of this pathway can be referred to as kynurenines, and the final product is the nicotinamide adenosine dinucleotide (NAD+) [2,25,27]. Proper regulation of the KP is crucial for the normal functioning of the vascular system and immune system to prevent autoimmune reactions [28].

Recent research focused on how the KP is chronically activated in neuroinflammatory and neurodegenerative states [25,29,30]. There is growing evidence that the majority of KP metabolites play a role in the pathogenesis of several neuropsychiatric diseases, including PD, schizophrenia, major depressive disorder (MDD), bipolar disorder, Huntington’s disease, multiple sclerosis, and AD, particularly when the KP becomes dysregulated [25,29,30,31,32,33,34,35,36,37,38,39,40,41,42,43,44,45,46,47,48].

In the first step of the pathway, TRP is oxidized by the cleavage of the indole ring, and formyl kynurenine is generated. This reaction is catalyzed by either tryptophan 2,3-dioxygenase (TDO), which resides primarily in the liver, or indoleamine 2,3-dioxygenase 1 (IDO-1) or IDO-2, which resides in macrophages, microglia, neurons, and astrocytes [49,50,51,52,53,54,55,56]. IDO-1 is upregulated by specific cytokines and proinflammatory molecules such as lipopolysaccharides (LPS), beta-amyloid peptides, and human immunodeficiency virus proteins [50,57,58]. The most potent stimulant has been found to be interferon-gamma (INF-γ) [59,60]. Formyl kynurenine is converted to the first stable intermediate of the pathway, kynurenine (KYN), through a reaction catalyzed by kynurenine formamidase. Kynurenine is transformed to kynurenic acid (KYNA) by kynurenine aminotransferase (KAT) [25,27]. KYN in turn can be metabolized to 3-hydroxykynurenine (3-HK) by kynurenine 3-monooxygenase (KMO). The three intermediates downstream of 3-HK, 3-hydroxyanthranilic acid (3-HAA), quinolinic acid (QUIN), and picolinic acid (PIC) can act as neuroprotective or neurotoxic compounds [27]. KYNA and PIC are neuroprotective metabolites, whereas 3-HAA and 3-HK are neurotoxic. In physiologic conditions, the KP favors the production of KYNA, PIC, or NAD+. However, the KP shifts to overproduction of QUIN and other neurotoxic molecules in inflammatory states [61]. The KP is shown in Figure 1.

#### Kynurenine Pathway Metabolites

QUIN is the most potent intermediate product of the KP and can induce neuronal death and chronic neuronal dysfunction [53,62,63,64,65,66,67]. Under physiologic conditions, QUIN (<100 nM) leads to an increase in NAD+ and stem cell proliferation [68,69]. Under inflammatory conditions in the brain, infiltrating macrophages, microglia, and dendritic cells are primary sources of QUIN production. Astrocytes uptake QUIN and catabolize it to NAD+ [27]. However, under conditions of high inflammation, this system can become saturated, and QUIN accumulates in the cells to reach toxic levels [27]. QUIN toxicity is enhanced in the brain regions where NMDA receptors are most prevalent, such as the hippocampus and striatum [70].

There are a variety of mechanisms through which QUIN is toxic. As an agonist of the NMDA receptor, QUIN can increase glutamate release by neurons, inhibit its uptake by astrocytes, and alter conversion to glutamine by astroglia’s glutamine synthetase, [71] leading to neuronal excitotoxicity induced by excessive glutamate concentrations [72,73,74]. QUIN also potentiates glutamate toxicity and other excitotoxins, such as NMDA, by producing progressive mitochondrial dysfunction and subsequent energy depletion [75,76,77,78]. It also forms a complex with iron that transfers electrons to oxygen, creating reactive oxygen species (ROS), which mediates lipid peroxidation [79,80]. QUIN disrupts the integrity of the BBB, causes morphologic and ultrastructural alterations in neurons that promote excitability, destroys cholinergic projections to the cortex, and significantly decreases Ach release and cholinergic neurotransmission in the brain [81,82,83,84,85].

KYNA acting as an antagonist at the glycine site of the NMDA receptor can also antagonize the effects of excitotoxins such as QUIN [86,87]. However, in disease states where QUIN production is high, there may be insufficient KYNA to block QUIN [75]. KYNA plays an antioxidant role, as it can scavenge several free radicals and protect against their toxic effects [88]. KYNA also acts as an endogenous ligand of G-protein-coupled receptor (GPR35), which inhibits LPS-mediated TNF-α release from activated forms of macrophages [89]. Therefore, KYNA may limit the cascade elicited by inflammatory mediators that induce IDO activity [89]. KYNA also serves as an antagonist (indirect) to α7nAChR [90,91,92]. Although KYNA is considered a neuroprotective molecule, this activity may impair the long-term potentiation and cognitive decline in PD-MCI and PDD [93,94].

3-HK is another neurotoxic intermediate of TRP metabolism [95]. It generates free radicals, such as superoxide and hydrogen peroxide, promoting copper-dependent oxidative protein damage leading to neuronal apoptosis and neurodegeneration [96]. 3-HK potentiates the excitotoxic and oxidative stress induced by QUIN [97] and impairs mitochondrial function [98]. 3-HAA is another TRP metabolite that acts as a free radical generator in the presence of copper [96], and intracerebral injections of 3-HAA decreased choline acetyltransferase activity [27]. Conversely, 3-HAA is also shown to have neuroprotective actions, as it can serve as an antioxidant by scavenging peroxyl radicals [99]. It also inhibits the activation of nuclear factor-kB and inducible nitric oxide synthase at low millimolar concentrations [100].

On the other hand, PIC, a known chelator of metal ions, is a neuroprotective compound within the brain [101]. It is believed that it controls cellular growth and has antitumoral, antifungal, and antiviral activities [102]. It is thought that PIC decreases kainate and calcium-induced glutamic acid release, providing neuroprotection to cholinergic neurons of the nucleus basalis magno-cellularis and the NAD diaphorase-containing striatal neurons of mice against QUIN neurotoxicity. Some evidence also suggests that PIC acts as a glycine agonist at strychnine-sensitive receptors [89]. Figure 2 illustrates these KP metabolites and their effects on the brain.

### 2.2. The Association Between the Kynurenine Pathway and Parkinson’s Disease

A growing body of evidence suggests that KP activation is involved with the neuropathology and pathogenesis of PD-MCI and PDD [39,103]. Below, we describe this kynurenine acid pathophysiology model, as shown in Figure 3.

The KP was first associated with PD in 1992 when Ogawa and colleagues found significantly reduced concentrations of KYN and KYNA in the frontal cortex and increased levels of 3-HK in the putamen and substantia nigra pars compacta (SNpc) of PD patient tissue [103]. Another study found decreased KYNA in PD patients’ cortical areas, caudate, and cerebellum [75]. KAT expression is also decreased in the SNpc of MPTP-treated mice [104]. Several studies have revealed significant reductions or elevations of KP metabolites in the urine, serum, or cerebrospinal fluid (CSF) of PD patients as compared to healthy controls (HCs) [105,106,107,108]. Specifically, KYN/TRP ratio is increased in the CSF and serum of PD patients compared to HCs [109,110]. Increased KYN was also reported in the urine of PD patients in two additional studies [105,106]. Together, these findings suggest increased KP activation in the brains of PD patients. KAT I and KAT II activities, along with plasma KYNA, were found to be significantly lower in the plasma of PD patients [107]. An analysis (N = 48 PD; N = 57 HCs) found a 33% increase in 3-HK concentration in the CSF of PD patients compared to HCs [108]. A study examining altered KP metabolism in the plasma and CSF of PD patients with L-DOPA-induced dyskinesia also found a shift toward an increase in 3-HK and a decrease in KYNA [111]. Another study (N = 82 PD; N = 82 HCs) reported increased QUIN and decreased KYNA in the plasma of PD patients compared to HCs [112]. A third study (N = 33 PD; N = 39 HCs) found reduced KYNA plasma levels in the PD group. This study also found an association between aging and the accumulation of KYNA, QUIN, and KYN [39]. A final study (N = 20 PD; 13 HCs) found significantly higher levels of KYN and 3-HK in the CSF of the PD cohort compared to that of HCs. This study did not see a difference in CSF levels of KYNA or QUIN but did find significantly higher levels of TNF-α and IL-1α in the PD cohort. These cytokines have been connected to activation of KMO [113].

#### 2.2.1. α-Synuclein Aggregation

Recently, chronic intestinal inflammation, alterations in the gut microbiome, and the spreading of αSyn aggregates from the gut to the brain via the vagal nerve have been linked to PD pathogenesis [114,115]. The KP is implicated in this process, as microbiome studies show that the gut microbiota influences it [116]. Under normal physiologic conditions, the KP is used in the gut to prevent or alleviate intestinal inflammation [29]. However, it has been shown that excessive levels of QUIN due to KP dysregulation led to the formation of metabolite assemblies that cause αSyn aggregation [117,118]. Another study supported this view, as inoculation of mice with QUIN resulted in increased levels of phosphorylated αSyn [118].

Low-grade inflammation, mitochondrial dysfunction, and oxidative stress are strongly associated with aging, the greatest known risk factor for PD [6,7,8,119]. Inflammatory mediators like IFN-γ, TNFα, toll-like receptors (TLRs) 1–6 and 9, LPS, and amyloid activate the KP through IDO-1 [58,120,121]. KYNA is also involved in leukocyte recruitment and may mediate anti-inflammatory effects in the brain [122]. Therefore, low-grade inflammation, aging, and KP activation are all linked together in their involvement in the pathophysiology of PD [123,124,125].

#### 2.2.2. Neurotoxicity

KYNA functions as a mediator of neuroprotection in PD through its antagonism of the NMDA receptor and by slowing down the excitotoxic cascade in neuronal cells, which has been demonstrated in clinical studies [75]. Low levels of endogenous KYNA decrease its ability to limit the excitotoxicity induced by high levels of QUIN or glutamate through the NMDA receptors [75]. During immune activation, TRP metabolism can become unbalanced, and it is catabolized through the KP instead of the 5-HT/melatonin pathway. This leads to a decrease in 5-HT, which may contribute to depressive symptoms during PD progression [119]. There is evidence that QUIN plays a role in the depressive symptoms seen in PD. It is increased in the CSF of patients with depression and suicidal thoughts indicating that it may play have a role in the generation of depressed mood [45,46,47,48]. KYNA has also been found to be decreased in the CSF of suicidal patients [46].

## 3. Galantamine and Memantine

### 3.1. Preclinical Evidence

Galantamine is an AChEI is a positive allosteric modulator (PAM) of presynaptic α7 nicotinic acetylcholine receptors (α7nAChR) as well as the α4β2 subtype of nicotinic cholinergic receptors, the most abundant nicotinic receptor in the brain [126]. Galantamine’s efficacy has been studied extensively in AD, and it is documented that chronic administration improves cognitive function and delays the development of behavioral changes associated with the disease [25,127,128,129,130,131,132,133,134]. Galantamine is approved by the FDA for the treatment of moderate to severe AD [23]. However, it has been used off-label to treat a variety of related disorders such as vascular dementia, mixed dementia, PD, dementia with Lewy bodies (DLB), frontotemporal dementia, and dementia associated with multiple sclerosis [135]. There are many proposed mechanisms for improving cognitive which include some of the mechanisms used by medication to treat dementia. The α7nAChR action of these medications facilitates the release of ACh from the presynaptic neurons [126]. nAChRs in the CNS are expressed on the presynaptic neuronal membrane and control the release of major neurotransmitters such as ACh, GABA, glutamate, norepinephrine, DA, and serotonin [126]. Studies show that agonists of nAChRs improve cognitive functions, while antagonists of nAChRs cause impairment of cognitive processes [136,137]. Among AChEIs, galantamine may be the superior choice compared to donepezil and rivastigmine because of this dual action [138]. Furthermore, galantamine improves α-amino-3-hydroxy-5-methyl-4-isoxazolpropionate (AMPA)-mediated signaling, which could be neuroprotective and may improve memory coding [139]. Galantamine also has an antiapoptotic effect as a scavenger of ROS [140,141,142,143,144].

Memantine is an antagonist of the NMDA subtype of the glutamate receptor [145]. More specifically, memantine is an uncompetitive, low-affinity, open-channel blocker that preferentially enters the receptor-associated ion channel only when it is excessively open thus preventing memantine from interfering with normal synaptic transmission [145]. Also, it allows memantine to protect against neuronal damage and death induced by glutamate excitotoxicity [145]. Glutamate excitotoxicity is relevant in various acute and chronic neurologic disorders including dementia [146]. Memantine enhances the elimination of damaged mitochondria in neuronal models and has antioxidant activity [147]. This may be beneficial for the treatment of PD, as abnormal mitophagy has been implicated in PD pathogenesis [148]. The effectiveness of memantine has been studied extensively in AD, and some meta-analyses favor it as a first-line anti-dementia treatment [147]. These analyses also suggest that the combination of memantine and donepezil (AChEI) is safe and can provide further benefits [148,149]. Memantine is currently FDA approved for the treatment of moderate to severe AD, but it has also been used off-label for the treatment of MCI, mild AD, vascular dementia, chronic pain, and psychiatric disorders [24,145,150].

The combination of both medications is intriguing, and the α7 nicotinic-NMDA hypothesis has been recently published [44,94,151]. Initially, it appears that the two drugs act in an opposing manner. Galantamine increases glutamate release through the presynaptic α7nAChR, whereas memantine decreases glutamate release by acting as an NMDA antagonist [126,145]. Galantamine is an AChEI and may have a dual action, as it has been reported to be a positive allosteric modulator (PAM) of α7 nicotinic acetylcholine receptors (α7nAChR) as well as the α4β2 subtype of nicotinic cholinergic receptors, which are the most abundant nicotinic receptor in the brain [126]. However, the original report of PAM activity using rodent tissue and cell lines has not been repeated, and galantamine has no PAM activity on nicotinic receptors on human cells [152]. It is possible that the advantage of galantamine over other cholinesterase inhibitors is due to its potentiation of NMDA receptors [153]. It is also important to note that it is unlikely that a treatment using the combination memantine and galantamine would affect the metabolism of either drug. Galantamine is metabolized by cytochrome P450 (CYP) 2D6 and CYP3A4, neither of which are affected by memantine [154]. The use of this combination is also supported by pharmacodynamic and pharmacokinetic studies [155,156,157,158].

The synergistic improvement in cognition using the galantamine-memantine combination has been demonstrated [159]. It has been shown in mice that rescue of scopolamine-induced memory impairment is possible with low sub-active doses of galantamine and memantine [160]. Additionally, NMDA toxicity has been prevented in rat cortical neurons with memantine and galantamine [161]. The memantine-galantamine combination also improves cognitive performance in nonhuman primates (aged rhesus macaques) and has facilitated rat attentional set-shifting task performance and reversed delay-induced deficits in object recognition [160,162,163]. This drug combination may also provide greater cognitive benefits than either medication alone [164,165]. A single antioxidant may be inadequate to counteract the complex cascade of redox state dysfunction triggered by neurodegenerative disorders such as PD [166]. Double antioxidant treatment with the galantamine-memantine combination has been proposed in schizophrenia [151] and may also be relevant in PD.

### 3.2. Galantamine-Memantine Combination and Kynurenine Pathway

As stated previously, KYNA serves as an NMDA antagonist which limits excitotoxic stimulation of NMDA receptors by the neurotoxic KP metabolite QUIN [25,29,43,86,119]. Memantine can achieve the same antioxidant mechanism. Although KYNA is recognized as a neuroprotectant, it is also known for its inhibitory actions on cholinergic transmission in the brain [91,167]. KYNA blocks the effects of α7nACh receptors indirectly [90,168]. The ability of galantamine to stimulate α7nAChR and other nicotinic receptors, including α4β2, could potentially counteract the negative actions of KYNA [169]. Using galantamine and memantine in combination allows for both antioxidant while also negating the negative actions of KYNA.

Stimulation of the cholinergic system downregulates the inflammatory immune response, which is known as the cholinergic anti-inflammatory pathway [170]. Therefore, α7nAChR antagonism of KYNA may contribute to inhibiting the cholinergic anti-inflammatory pathway. Early developmental elevations of brain KYNA are associated with cognitive impairments in adult rats, which are reversed with galantamine [169].

### 3.3. Galantamine and Memantine: Clinical Evidence

Galantamine is considered by the International Parkinson and Movement Disorder Society Evidence-Based Medicine Review to be potentially useful as an intervention in treating PDD [2]. An open-label trial of galantamine at a maximum dose of 16 mg/day that included participants with PDD (N = 21, galantamine; N = 20, HC) showed statistically significant improvements in the Mini-Mental State Examination (MMSE), the Cognitive Alzheimer’s Disease Assessment Scale (ADAS-cog), the clock drawing test, and the Frontal Assessment Battery (FAB). Galantamine-treated participants also showed benefits in the Neuropsychiatric Inventory, improving symptoms such as hallucinations, anxiety, sleep disturbance, and apathy [171].

Memantine is also considered by the International Parkinson and Movement Disorder Society Evidence-Based Medicine Review to be a potentially useful as an intervention in treating PDD [2]. One study evaluated the cognitive effects of memantine in DLB and PDD using automated tests of attention and episodic memory. Thirty PDD patients were included in the study, and memantine produced statistically significant improvement with an effect size ranging from 0.75 to 0.79 in choice reaction time and immediate and delayed word recognition (3/4 tasks), respectively [172]. However, a meta-analysis that included three studies, one randomized controlled trial (RCT) specific for PDD and two for mixed DLB and PDD, showed no significant differences between groups in health improvement or absence of deterioration [173]. This underscores the need to conduct RCTs with the combination.

The galantamine-memantine combination has not yet been studied in an RCT for treating PD-MCI or PDD. However, it may be beneficial in treating these conditions because of their individual beneficial actions and the potential for synergistic effects. Current evidence suggests that the galantamine-memantine combination could modulate the negative effects of KP metabolites on cognition [174]. PD, like other neurodegenerative diseases, involves multiple pathophysiologic mechanisms. Combination treatment seems to be a logical approach, as individual drugs may only target one of a few pathways involved in the disease [175,176,177]. Combination treatment has been proposed for use in AD, and unsuccessful trials with individual drugs should not be a barrier to their use in a combination that engages a wider range of therapeutic targets [175]. Currently, only two pathophysiologic mechanisms (cholinergic/nicotinic-cholinergic and glutamatergic/NMDA) have been approved by the FDA for the treatment of cognitive dysfunction in AD [156,178]. As the pathophysiologic mechanisms of PDD are similar, we extrapolate this logic to apply to PDD and PD-MCI as well. Although this combination is the standard of care in AD, the use of this combination would be a novel treatment in PDD [179].

Data from a two-year RCT showed significant cognitive benefit for the prodromal stage of AD treated with the galantamine-memantine combination compared with galantamine alone. This study also found that cognitive decline occurred after galantamine was discontinued [180]. In a naturalistic study, patients diagnosed with DLB were treated with galantamine for six months, with those responding (19/38) also administered memantine. Using this study design, the addition of memantine significantly improved cognition compared to galantamine on its own [135]. RCTs that studied the combination of AChEIs and memantine in AD have shown a slowed cognitive decline and improved cognition compared to AChEI monotherapy [181,182,183]. Finally, a retrospective cohort study of AD showed that the galantamine-memantine combination (N = 53) significantly improved cognition and apathy compared to the donepezil-memantine combination (N = 61) [184].

An interactive effect of the galantamine-memantine combination in reducing KP metabolites has been documented extensively [151]. The combination of galantamine and memantine can modulate KP metabolites, resulting in cognitive enhancement [174]. In an open-label study with two participants with schizophrenia treated with the galantamine-memantine combination for six weeks, cognitive battery scores and KP metabolites in plasma were investigated. In the first participant, the treatment improved cognitive scores in processing, speed, attention/vigilance, visual learning, reasoning, problem-solving, and social cognition. There were improvements in speed of processing and working memory scores in the second participant. The study also found decreased TRP, PIC, KYN, and KYNA concentrations with treatment. These are novel biomarkers that can be used to monitor progress with treatment [185]. Therefore, this combination may be a future treatment to improve cognition in patients with PDD.

Furthermore, disruption in the KP has been implicated in psychotic and negative symptoms in schizophrenia, MDD, bipolar disorder, and other neurologic conditions including traumatic brain injury [45,46,186,187,188]. Sixty percent of patients with PD who have been treated with carbidopa-levodopa report experiencing intermittent psychotic symptoms [189]. Also, 20–40% of patients with PD progress to a continuous psychotic state as their disease course progresses [190]. As patients with PD can suffer from psychotic, depressive, or other symptoms overlapping these disorders, targeting the KP also has the potential to benefit patients in these areas, in addition to cognition [191]. Both galantamine, due to its action on α7nAChR, and memantine, due to its action on NMDA, are likely to be effective in treating depressive symptoms as well [192,193,194,195,196].

Many patients with PD experience apathy, which is defined as a lack of motivation not attributable to diminished levels of consciousness, cognitive impairments, or emotional distress [197]. Apathy in this case is more likely due to a combination of negative symptoms such as impaired motivation, drive, initiation, and emotional reactivity [198]. These negative symptoms are not unique to PD and are found in other neurologic and psychiatric conditions, such as other forms of dementia and schizophrenia [199]. It may be worthwhile to consider using the galantamine-memantine combination to treat these other conditions as well [184].

## 4. Biomarkers

Biomarkers have wide utility and may aid in assessing risk, onset, progression, and efficacy in the treatment of PD. They may also become useful in a subclinical, preventative setting, as we know that most of the dopaminergic neurons in the SNpc are functioning at full capacity by the time of a PD diagnosis [32]. Examples of potential blood-based markers that have been identified for PD include apolipoprotein A1 (ApoaA1), uric acid, and epidermal growth factor (EGF) [200,201,202]. Low levels of plasma ApoaA1 correlate with increased risk for PD [201]. Low levels of uric acid and EGF correlate with PD motor symptoms and nonmotor symptoms (cognitive impairments), respectively [200,202,203].

Intermediates of the KP (KYNA, QUIN, 3-HK, PIC) have the potential to be revealing biomarkers for PD. Several studies show differences in KP intermediate levels in symptomatic patients and controls. They also show correlations between KP intermediate levels and phenotypic expression of the disease. KYN has been found to be elevated in the urine, blood, and CSF of patients with PD compared to controls [105,106,109,110,113]. QUIN is elevated in the blood [112]. 3-HK has been found to be elevated in blood and CSF [108,111,113]. KYNA has been shown to be reduced in the blood and CSF [39,107,111,112]. KP enzyme activity is another potential PD biomarker. IDO activity, as reflected by the KYN/TRP ratio, is upregulated in the blood of PD patients compared to controls [204]. KAT is reduced in the blood of PD patients and in MPTP-treated mice [104,107]. These metabolites modulate the immune system as well as neuron excitation and inhibition. Changes in the level of activity of the KP and balance of metabolites can also affect other routes of TRP metabolism and energy production. Moreover, the galantamine-memantine combination may also enhance mismatch negativity, brain-derived neurotrophic factor, and synaptic density, which are important in learning and memory processes [25,169,174,205]. In our opinion, they are not the main clinical objective at this time. After or in parallel to advancing new therapies, further research into the biomarkers discussed here and others has the potential to uncover future clinically useful diagnostic or prognostic markers.

## 5. Neuromodulation

With limited pharmaceutical options available to successfully treat PDD, the field has begun to explore the use of neuromodulation, including deep brain stimulation (DBS), transcranial magnetic stimulation (TMS), transcranial direct or alternating stimulation (TDAS), and focused ultrasound [206]. By far the most used of these interventions is DBS. In 2018, over 100,000 patients worldwide had been treated for PD using DBS [207]. DBS works by providing electrical stimulation to the subthalamic nucleus or globus pallidus interna via surgically implanted electrodes [208]. Although helpful in treating motor symptoms for most patients, they may not respond any better than they do to levodopa and therefore will still require medication. Currently, DBS is contraindicated in patients with cognitive impairments or severe psychiatric disease, both of which can worsen with this treatment [209]. DBS also carries risk of hemorrhage, stroke, infection, and death [208]. Other forms of neuromodulation are noninvasive and may be better options in PD patients for whom cognitive status is a concern. The second most common neuromodulation technique used in the treatment of PD is TMS, which targets the dorsolateral prefrontal cortex (DLPFC), a brain region that plays a key role in a wide variety of cognitive functions and higher-level processing [206,210]. A study showed that undergoing two weeks of 15Hz TMS targeting the DLPFC improved performance on a Stroop Task Test, which is used to assess the ability to inhibit cognitive interference that occurs when the processing of a specific stimulus feature impedes the simultaneous processing of a second stimulus [211]. The study consisted of two arms, one receiving TMS (N = 13) for 2 weeks and the other receiving fluoxetine (N = 12, 25 mg per day) for 8 weeks. Both groups showed similar improvements [212]. This study had no control group and a very small sample size, making it difficult to truly discern whether or not TMS improves cognition in PD patients [212]. Similarly to DBS and TMS, all other forms of neuromodulation have not been found to be consistently effective in treating PDD [206]. Unlike the proposed combination of galantamine and memantine, neuromodulation demands higher effort and cost from the patients. All these interventions require frequent office visits and even hospitalization. A systematic review estimated the average cost of DBS to be $186,244 over five years, which is more than the best medical treatment [213]. When adjusted for growing inflation, this cost is unacceptable to most patients. For comparison, 60 tablets of galantamine 8 mg cost only $171.80 [214]. Fifty tablets of memantine 10 mg costs only $32.43 [215]. This means that over the course of 5 years, the cost of this drug combination is $6409.28. Before factoring in insurance, which is far more likely to cover the cost of the galantamine-memantine combination, this saves patients $179,834.72 over a period of five years. These calculations suggest that neuromodulation is not a feasible answer to treating PDD in most of our population.

## 6. Why Have We Failed to Move the Needle?

Given the complexity of a disease with multiple pathways degenerating over time, it may be unrealistic to expect that targeting a single pathway could significantly affect cognition. Complex diseases often evolve, altering their mechanisms to resist the body’s natural response and to evade drug therapies. This concept is studied widely in cancers and cardiovascular diseases but may also apply to neurologic disorders such as PD and AD.

Prior research shows that using varying combinations of drugs at different doses can potentially modulate as many as 88% of relevant biochemical pathways [156,216]. Using combinations not only targets more than one pathway but also may allow for drugs to work synergistically at lower and safer doses [216]. Repurposing existing drugs at lower doses removes some of the need for continued novel drug discovery. In a study of new therapeutic drugs approved by the FDA between 2009 and 2018, the median research and development investment required to bring a new drug to market was estimated to be $985.3 million, and this figure will only continue to climb [217]. By using preexisting drugs in new combinations, the cost to the healthcare industry may be drastically lowered.

## 7. Other Combinations

Combination treatments are proposed for the treatment of a variety of diseases that are influenced by multiple pathophysiologic processes, including AD and schizophrenia [94,175]. The idea behind these proposed combinations is an impetus for the proposed galantamine-memantine combination: using multiple drugs that each target different pathophysiologic pathways in combination should produce better efficacy than any one drug could on its own. Further, the failure of one of the drugs to have significant efficacy on its own does not preclude its potential as part of a combination treatment [175]. AD and schizophrenia share certain underlying pathophysiologic pathways with PD, including KP, cholinergic, and glutamatergic alterations. In schizophrenia, drug combinations have been proposed to include an antipsychotic, one drug targeting glutamatergic/NMDAergic pathways, and another drug targeting cholinergic-nicotinic pathways [25]. The drugs targeting glutamatergic/NMDAergic pathways that can be used include memantine, *N*-acetylcysteine (NAC), D-cycloserine, glycine, sarcosine, D-serine, and acamprosate [218,219]. The drugs targeting cholinergic-nicotinic pathways include galantamine, cotinine, and varenicline [94,220]. An advantage of this approach is that drug choice can reflect an individual patient’s tolerability to specific drugs. As galantamine and cotinine are PAMs, while varenicline is an agonist of α7nAChR, galantamine or other PAMs like cotinine are more likely to be effective by avoiding agonist-induced desensitization by engaging the allosteric site of the α7nAChR [193,221]. Various combinations proposed based on this concept include antipsychotic-NAC-varenicline, antipsychotic-galantamine-NAC, and antipsychotic-galantamine-memantine [25,151,185]. All these combinations, like the galantamine-memantine combination for PDD, require RCTs to prove their efficacy to be used for the benefit of patients (222, 223). With combinations currently approved (donepezil-memantine) for treating AD and used off-label (galantamine-memantine), it is not unreasonable to propose that a combination is the answer to a better life for PDD patients (Table 1).

Additionally, for patients who remain symptomatic on the galantamine-memantine combination, adding NAC may provide further improvement in symptoms [151,218,219,224]. Further research on off-label uses and RCTs focused on galantamine-memantine, and other combinations are the next logical steps (Figure 4) [225].

## 8. Conclusions and Future Directions

Based on preclinical evidence, clinical trial data, and rationale, the use of galantamine and memantine as a combination may be more effective than if used individually. Use of this combination would not only lower healthcare costs by reducing the need for spending on drug development, but, more importantly, it could improve patients’ quality of life. Future RCTs are needed to prove the efficacy of the galantamine-memantine combination in PDD.

## Figures and Tables

**Figure 1 brainsci-14-01163-f001:**
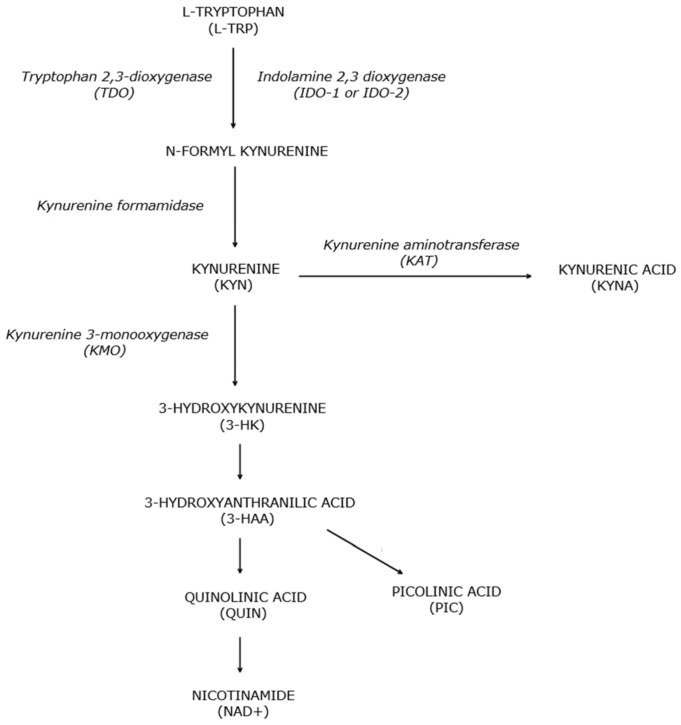
Kynurenine pathway. An abbreviated depiction of the kynurenine pathway showing the major steps.

**Figure 2 brainsci-14-01163-f002:**
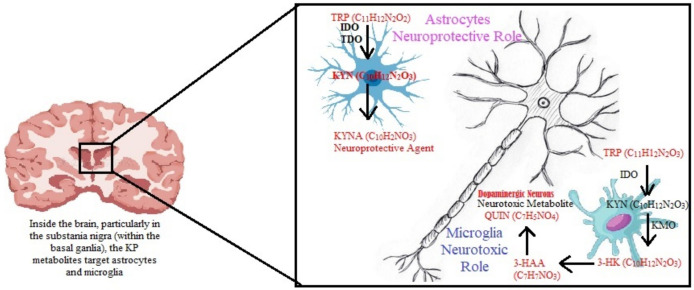
Overview of the kynurenine pathway in the brain and its effects. Depiction of the differential expression of the KP in cells of the central nervous system. Astrocytes lack the full complement of KP enzymes, hence KP activation in astrocytes terminates in the production of neuroprotective KYNA. However, as microglia fully express KP enzymes, KP activation in microglia can result in the production of neurotoxic metabolites 3-HK and QUIN. KP = Kynurenine Pathway; TRP = Tryptophan; KYNA = Kynurenic Acid; IDO = Indoleamine 2,3-dioxygenase; TDO = Tryptophan-2,3-dioxygenase; QUIN = Quinolinic Acid; 3-HAA = 3 Hydroxyanthranilic Acid; 3-HK = 3-hydroxykynurenine; KMO = Kynurenin-3-monooxygenase; KYN = Kynurenine.

**Figure 3 brainsci-14-01163-f003:**
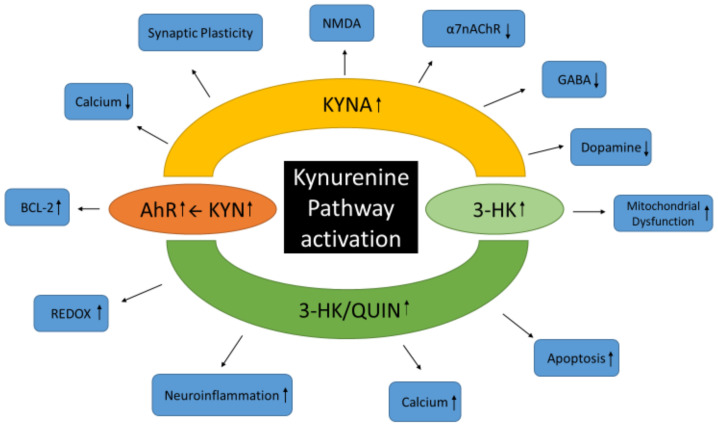
Kynurenine pathway-centric pathophysiology model. Depiction of some of the receptors, pathways, and processes affected by increased levels of major kynurenine pathway metabolites KYN, KYNA, 3-HK, and QUIN after pathway activation. AhR = aryl hydrocarbon receptor; α7nAChR = Alpha7 nicotinic receptor; BCL-2 = B-cell Lymphoma 2; GABA = γ-aminobutyric acid; KYN = Kynurenine; KYNA = Kynurenic Acid; NMDA = N-methyl-D-aspartate; QUIN = Quinolinic Acid; 3-HK = 3-hydroxykynurenine. ↑, increased process; ↓, decreased process.

**Figure 4 brainsci-14-01163-f004:**
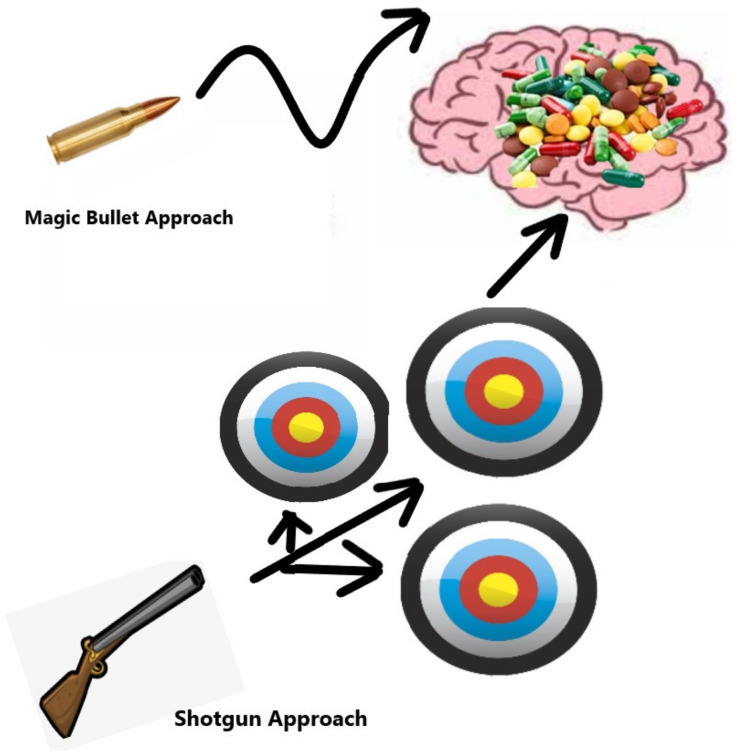
Magic Bullet Versus Shotgun Approach. The magic bullet approach has long been thought to be the answer to treating complex medical conditions. Pharmaceutical companies hoped that they would be able to develop a single drug to treat many conditions. However, this has failed countless times. We argue that the shotgun approach is more effective. Using multiple drugs (shotgun approach) to target multiple pathways implicated in a disease is likely to a more effective treatment approach [225].

**Table 1 brainsci-14-01163-t001:** Meta-Analyses of Randomized Controlled Trials in Schizophrenia: Potential Medications in Parkinson’s Disease (PD) ^a^.

Studies	Cognitive Symptoms	Positive Symptoms	Negative Symptoms
NMDAergic Drugs			
Zheng et al. (2019) Memantine Nine studies (N = 512) [222]	ES = 1.07, *p* < 0.0001	ES = 0.32, *p* = 0.05	ES = 0.71, *p* = 0.0003
Yolland et al. (2020) *N*-acetylcysteine Seven studies (N = 220) [223]	Working memory (ES = 0.56, *p* = 0.005). Processing speed ES = 0.27, Not Significant	ES = 0.21, *p* = 0.05	ES = 0.72, *p* = 0.003
Nicotinergic Drugs			
Koola et al. (2020) Galantamine Six studies (N = 226) [30]	ES = 0.233, *p* < 0.001 Five studies: ES = 0.269 (only studies with 24 mg)	ES = 0.076, Not significant	ES = 0.107, not significant
Tanzer et al. (2020) Varenicline Four studies (N = 339) [224]	ES = 0.022, Not significant	No data	No data

^a^ The effect sizes (ES) of four medications (memantine, galantamine, *N*-acetylcysteine, and varenicline) on cognitive symptoms, positive symptoms, and negative symptoms in schizophrenia studies are shown in this table. Because of the considerable overlap in the pathophysiology between schizophrenia and PD, these effect sizes may translate to PD as well.

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
