# Peer review of "Galantamine-Memantine Combination in the Treatment of Parkinson’s Disease Dementia"

_brainsci, 2024, doi:10.3390/brainsci14121163_

Round 1

Reviewer 1 Report

Comments and Suggestions for Authors

The manuscript titled: „Galantamine-Memantine Combination in the Treatment of Parkinson’s Disease Dementia” provides a comprehensive review of literature data in that area.

The first paragraph is Introduction, in which Authors characterized  Parkinson’s Disesase (PD) – its major symptoms, including PD dementia, risk factors and current possibilities of pharmacotherapy. They pointed out that galantamine and memantine were approved for cognitive disfunction in Alzheimer’s disease but not in PD.

The second paragraph presents kynurenic pathway as important pathway strongly associated with PD. Authors describ all steps in L- tryptophan metabolism and show it graphically. They point out metabolites acting as neuroprotective agents (KYNA, PIC) and metabolites which may be harmful for  neurons. They precisely describe the connections between kynurenic pathway and PD.

In the third paragraph Authors described in details galantamine and memantine – their mechanisms of action, also preclinical and clinical evidences  for its potential in therapy of PDD.

In the subsequent paragraphs the endogenous markers of kynurenic pathway are described which can be useful in monitoring of PD. Besides, other possibilities of PD therapy are also presented, such as, deep brain stimulation, transcranial magnetic stimulation. Additionally, various drug combinations are also mentioned.

·       There are many abbreviations in whole manuscript therefore I recommend to make a list of abbreviations.

·       I saw only one mistake – line 100 – change “cognition” into “cognitive disfunction”

Summing up, in my opinion that manuscript is valuable and well prepared. It collects a wide range of current data in innovative  posibbilities of PD therapy. In my opinion that manuscript can be accepted for publication.

Author Response

REVIEWER 1

The manuscript titled: „Galantamine-Memantine Combination in the Treatment of Parkinson’s Disease Dementia” provides a comprehensive review of literature data in that area. The first paragraph is Introduction, in which Authors characterized  Parkinson’s Disesase (PD) – its major symptoms, including PD dementia, risk factors and current possibilities of pharmacotherapy. They pointed out that galantamine and memantine were approved for cognitive disfunction in Alzheimer’s disease but not in PD.

The second paragraph presents kynurenic pathway as important pathway strongly associated with PD. Authors describ all steps in L- tryptophan metabolism and show it graphically. They point out metabolites acting as neuroprotective agents (KYNA, PIC) and metabolites which may be harmful for  neurons. They precisely describe the connections between kynurenic pathway and PD. In the third paragraph Authors described in details galantamine and memantine – their mechanisms of action, also preclinical and clinical evidences  for its potential in therapy of PDD. In the subsequent paragraphs the endogenous markers of kynurenic pathway are described which can be useful in monitoring of PD. Besides, other possibilities of PD therapy are also presented, such as, deep brain stimulation, transcranial magnetic stimulation. Additionally, various drug combinations are also mentioned. There are many abbreviations in whole manuscript therefore I recommend to make a list of abbreviations.

Authors: A list of the abbreviations was included in the end of the manuscript.

I saw only one mistake – line 100 – change “cognition” into “cognitive disfunction”

Authors: This change was corrected

Summing up, in my opinion that manuscript is valuable and well prepared. It collects a wide range of current data in innovative posibbilities of PD therapy. In my opinion that manuscript can be accepted for publication.

Authors: We appreciate the time to correct and improve the quality of our manuscript.

Reviewer 2 Report

Comments and Suggestions for Authors

A research paper that is well-written and has enough details to support the title. For readers and academics who are interested in this kind of research, it is regarded as a useful addition.

Author Response

REVIEWER 2

A research paper that is well-written and has enough details to support the title. For readers and academics who are interested in this kind of research, it is regarded as a useful addition.

Authors: Dear Reviewer we appreciate your comment, and time to improve the quality of the current manuscript.

Reviewer 3 Report

Comments and Suggestions for Authors

Congratulations on your good work.

The manuscript´s content is interesting and well-presented.

My small comments are:

- please make all titles in the manuscript without abbreviations

- write one paragraph about non-pharmacological treatment

- give some information about the importance of early diagnosis 

Thank you

Author Response

REVIEWER 3

Congratulations on your good work.

The manuscript´s content is interesting and well-presented.

My small comments are:

please make all titles in the manuscript without abbreviations

Authors: Correction was made

write one paragraph about non-pharmacological treatment

Authors: We chose not to add an additional paragraph at this time as section 5 regarding neuromodulation already discussed non pharmacological options for treatment which unfortunately at this time are very limited.

give some information about the importance of early diagnosis

Authors: A few brief sentences were added to lines 66-69. It now reads “Additionally, rapid eye movement (REM) sleep disturbances and sensory abnormalities are an inherent part of the disease, which are often some of the first presenting symptoms [1]. Early detection and early treatment can significantly improve quality of life. Early detection allows medication to be started before these manifestations can lead to poor quality of life, hospitalization, and more significant health care costs to patients and caregivers [3, 4].”

Thank you

Authors: We appreciate the time of the Reviewer for correction and improvement of the quality of the manuscript.

Reviewer 4 Report

Comments and Suggestions for Authors

The manuscript “Galantamine-Memantine Combination in The Treatment of Parkinson’s Disease Dementia” is quite clear and well-presented and represents a good starting point for possible further studies on the efficacy of combined known molecules in the treatment of complex neuropsychiatric disorders. Repurposing existing drugs can be more effective than researching new molecules, yielding faster and more cost-effective results Therefore, a thorough review of the scientific literature is essential when proposing to test such combinations. However, a few minor interventions could be made to improve the understanding of the text and provide the reader with a more accurate grasp of the conclusions presented by the authors.

1.      A list of abbreviations could be useful.

2.      The paragraphs in lines 269-287, 334-345 could be improved to enhance comprehension.

Author Response

REVIEWER 4

The manuscript “Galantamine-Memantine Combination in The Treatment of Parkinson’s Disease Dementia” is quite clear and well-presented and represents a good starting point for possible further studies on the efficacy of combined known molecules in the treatment of complex neuropsychiatric disorders. Repurposing existing drugs can be more effective than researching new molecules, yielding faster and more cost-effective results Therefore, a thorough review of the scientific literature is essential when proposing to test such combinations. However, a few minor interventions could be made to improve the understanding of the text and provide the reader with a more accurate grasp of the conclusions presented by the authors.

A list of abbreviations could be useful.

Authors: A list of abbreviations was included in the end of the manuscript.

The paragraphs in lines 269-287, 334-345 could be improved to enhance comprehension.

Authors: Thank you providing this feedback. Lines 269-287 now read as “Some minor changes have been made so it now reads “ Galantamine is an AChEI is a positive allosteric modulator (PAM) of presynaptic α7 nicotinic acetylcholine receptors (α7nAChR) as well as the α4β2 subtype of nicotinic cholinergic receptors, the most abundant nicotinic receptor in the brain [126]. Galantamine’s efficacy has been studied extensively in AD, and it is documented that chronic administration improves cognitive function and delayed the development of behavioral changes associated with the disease [25, 127-134]. Galantamine is approved by the FDA for the treatment of moderate to severe AD [23]. However, it has been used off-label to treat a variety of related disorders such as vascular dementia, mixed dementia, PD, dementia with Lewy bodies (DLB), frontotemporal dementia, and dementia associated with multiple sclerosis [135]. There are many proposed mechanisms for improving cognitive which include some of the mechanisms used by medication to treat dementia.The α7nAChR action of these medications facilitates the release of ACh from the presynaptic neurons [126]. nAChRs in the CNS are predominantly expressed on the presynaptic neuronal membrane and control the release of major neurotransmitters such as ACh, GABA, glutamate, norepinephrine, DA, and serotonin [126]. Studies show that agonists of nAChRs improve cognitive functions, while antagonists of nAChRs cause impairment of cognitive processes [136, 137]. Among AChEIs, galantamine may be the superior choice compared to donepezil and rivastigmine because of this dual action [138]. Furthermore, galantamine improves α-amino-3-hydroxy-5-methyl-4-isoxazolpropionate (AMPA)–mediated signaling, which could be neuroprotective and may improve memory coding [139]. Galantamine also has an antiapoptotic effect as a scavenger of ROS [140-144].”

Lines 334-345 have bee rephrased to read as “As stated previously, KYNA serves as an NMDA antagonist which limits excitotoxic stimulation of NMDA receptors by the neurotoxic KP metabolite QUIN [25, 29, 43, 86, 119]. Memantine can achieve this same antioxidant mechanism. Although KYNA is recognized as a neuroprotectant, it is also known for its inhibitory actions on cholinergic transmission in the brain [91, 167]. KYNA blocks the effects of α7nACh receptors indirectly [90, 168]. The ability of galantamine to stimulate α7nAChR and other nicotinic receptors, including α4β2, could potentially counteract the negative actions of KYNA [169]. Using galantamine and memantine in combination allows for both antioxidant while also negating the negative actions of KYNA.

Stimulation of the cholinergic system downregulates the inflammatory immune response, which is known as the cholinergic anti-inflammatory pathway [170]. Therefore, α7nAChR antagonism of KYNA may contribute to inhibiting the cholinergic anti-inflammatory pathway. Early developmental elevations of brain KYNA are associated with cognitive impairments in adult rats, which are reversed with galantamine [169].”